# Adoptive Transfer of CX3CR1-Transduced Tregs Homing to the Forebrain in Lipopolysaccharide-Induced Neuroinflammation and 3xTg Alzheimer’s Disease Models

**DOI:** 10.3390/ijms252413682

**Published:** 2024-12-21

**Authors:** Hyejin Yang, Juwon Yang, Namgyeong Park, Deok-Sang Hwang, Seon-Young Park, Soyoung Kim, Hyunsu Bae

**Affiliations:** 1Department of Physiology, College of Korean Medicine, Kyung Hee University, 26 Kyungheedae-ro, Dongdaemoon-gu, Seoul 02447, Republic of Korea; emilly86@naver.com (H.Y.); jwyang@khu.ac.kr (J.Y.); psys12@naver.com (S.-Y.P.); samanda0@nate.com (S.K.); 2Department of Clinical Korean Medicine, Graduate School, Kyung Hee University, 26 Kyungheedae-ro, Dongdaemoon-gu, Seoul 02447, Republic of Korea; moonholic36@gmail.com (N.P.); soulhus@khu.ac.kr (D.-S.H.)

**Keywords:** microglia, regulatory T cell, neurodegeneration, lipopolysaccharide, neuroinflammation, 3xTg mouse model, Alzheimer’s disease

## Abstract

CX3CR1-transduced regulatory T cells (Tregs) have shown potential in reducing neuroinflammation by targeting microglial activation. Reactive microglia are implicated in neurological disorders, and CX3CR1-CX3CL1 signaling modulates microglial activity. The ability of CX3CR1-transduced Tregs to inhibit LPS-induced neuroinflammation was assessed in animal models. CX3CR1 Tregs were administered to LPS-induced and 3xTg Alzheimer’s mouse models, resulting in reduced proinflammatory marker expression in both the cortices and hippocampi. In the 3xTg Alzheimer’s model, neuroinflammation was significantly reduced, demonstrating the efficacy of CX3CR1 Tregs even in chronic neuroinflammatory conditions. These findings highlight the therapeutic potential of CX3CR1 Treg therapy in modulating microglial activity and offer promising treatment strategies for neurodegenerative diseases.

## 1. Introduction

Alzheimer’s disease (AD) is a leading cause of neurodegenerative disorders characterized by dementia and the progressive loss of neurons, with the number of deaths increasing each year, making it a significant global public health issue [1]. In recent years, disease-modifying therapies such as Aducanumab and Lecanemab have been developed, focusing on reducing amyloid-β plaque accumulation. However, these treatments have shown limited effectiveness in slowing cognitive decline, and the results from clinical trials have sparked controversy. This highlights the need for a more comprehensive approach targeting other pathological mechanisms, such as tau proteins and inflammation [2].

In neurodegenerative diseases, including AD, a specific subtype of microglia called disease-associated microglia (DAM) is found [3,4]. Microglia, which belong to the innate immune system in the central nervous system (CNS), are key to interpreting and propagating inflammatory signals [5,6,7]. Activated microglia with proinflammatory phenotypes upregulate the expression of inflammatory mediators such as inducible nitric oxide synthase (iNOS) and TNF-α, accelerating AD pathology [8,9,10]. Cytokines such as TNF-α activate other immune cells, contributing to the development of neurodegenerative disorders [11]. Microglial activity can be modulated by factors such as fractalkine (CX3CL1) [12].

Recent studies have identified the CX3CL1-CX3CR1 interaction as a critical mechanism in microglial regulation. CX3CL1, which is highly expressed in the brain, interacts with the fractalkine receptor (CX3CR1) on microglia to regulate their activation, suggesting a unique communication system between neurons and microglia [13]. Furthermore, recently developed CX3CR1 inhibitors have shown that delayed administration of CX3CL1 can suppress proinflammatory microglia, indicating that CX3CR1 signaling may even inhibit reactive microglia [14,15,16].

Regulatory T cells (Tregs) are crucial for suppressing autoimmune responses and maintaining immune homeostasis. These cells inhibit the production of inflammatory cytokines, modulate immune activity, and exert immune suppression, thereby preventing excessive immune responses and slowing the progression of inflammatory diseases [17]. In the CNS, Tregs also play a crucial role in suppressing neuroinflammation. In AD and other neurodegenerative disorders, Tregs have been shown to inhibit the overactivation of microglia, providing neuroprotective effects. Tregs reduce the expression of proinflammatory cytokines and regulate the proinflammatory activation of microglia, helping to control neuroinflammatory processes [18,19].

Our previous study demonstrated that the administration of mouse Tregs into 3xTg-AD mice had neuroprotective effects on AD [20]. Additionally, recent studies have shown that the administration of antigen-specific amyloid-beta (Aβ)-positive Tregs can attenuate the progression of AD, suggesting that the bystander effect of Tregs could serve as a potential therapeutic strategy [21,22].

CX3CR1-expressing Tregs (CX3CR1^+^ Tregs) have been studied for their ability to selectively migrate to sites of inflammation through the CX3CL1/CX3CR1 axis. In atherosclerosis models, CX3CR1^+^ Tregs were shown to effectively migrate to inflamed areas and suppress immune–inflammatory responses, suggesting their potential as a therapeutic strategy for other inflammatory conditions, including neuroinflammatory diseases [23].

In this study, we aimed to evaluate whether CX3CR1^+^ Tregs, which have enhanced trafficking to inflammatory sites through the CX3CL1/CX3CR1 axis, can exert stronger anti-inflammatory and neuroprotective effects in AD. CX3CR1^+^ Tregs are hypothesized to migrate more effectively to inflamed areas, where they can suppress the activation of proinflammatory microglia. First, we assessed the ability of CX3CR1^+^ Tregs to suppress the expression of inflammatory cytokines in an LPS-induced neuroinflammation model, and we then evaluated their effect on chronic neuroinflammation and cognitive function in the 3xTg-AD mouse model. Therefore, this study focused on evaluating the potential of CX3CR1^+^ Tregs to modulate neuroinflammation and improve cognitive function in the context of AD pathology.

## 2. Results

### 2.1. CX3CR1 Retrovirus-Transduced Tregs Migrate to the Brain

To evaluate the effects of engineered Tregs on AD pathology, Tregs were transduced with a CX3CR1^+^ retrovirus. Control (CTRL) and CX3CR1^+^ Tregs were sorted and adoptively transferred to LPS-induced and 3xTg-AD mice (Figure 1A,B). To assess brain trafficking, CX3CR1^−^Flag^−^ and CX3CR1^+^Flag^+^ Tregs were generated from the splenocytes of Thy1.1 mice [24]. Five-month-old 3xTg-AD mice (Thy1.2^+^) received an adoptive transfer of 1 × 10^5^ Thy1.1^+^ Tregs. Seven days later, the mice were sacrificed, and the inguinal lymph nodes, kidneys, and brains were harvested. T cells were enriched from the kidneys and brains using 30–70% Percoll density gradient centrifugation and analyzed by flow cytometry. Flow cytometry analysis revealed that CX3CR1^+^Flag^+^ Tregs were significantly more prevalent in the brains (* *p* < 0.05), while no significant differences were observed between CX3CR1^−^Flag^−^ and CX3CR1^+^Flag^+^ Tregs in the inguinal lymph nodes or kidneys. These data suggest that CX3CR1 expression may enhance the migration of Tregs to the brain (Figure 1C,D).

### 2.2. CX3CR1^+^ Tregs Alleviate Cognitive Decline in LPS-Induced Neuroinflammation Mouse Model

Mice were divided into four groups, each containing five males. The control group received daily PBS injections and was also administered PBS as a vehicle control instead of Tregs. The LPS group was treated with LPS at a dose of 500 μg/kg, given daily for a week, and similarly received PBS injections instead of Tregs. The control Treg group and the CX3CR1 Treg group also received daily LPS injections like the LPS group. Additionally, 24 h after the last LPS injection, the control Treg group received Tregs without CX3CR1, and the CX3CR1 Treg group received Tregs with CX3CR1, both administered intravenously at 1 × 10^5^ cells per mouse. Cognitive function was evaluated using Y-maze and passive avoidance tests. The passive avoidance test is designed to evaluate the ability of animals to learn and avoid a painful stimulus. During training, mice learn that entering the dark compartment results in an electric shock, and this test is used to assess non-declarative memory. Mice treated with LPS and CTRL Tregs did not remember this association well, spending less time in the illuminated compartment. In other words, mice with impaired memory were more likely to enter the dark compartment despite the previous shock. In contrast, mice treated with CX3CR1^+^ Tregs showed improved memory, avoiding the dark compartment and spending more time in the illuminated area, indicating that CX3CR1^+^ Tregs improved memory and learning abilities (Figure 2B).

The Y-maze test is used to evaluate short-term spatial working memory in mice. Spontaneous alternation behavior, a measure of spatial working memory, is driven by the innate curiosity of rodents to explore new environments. Mice with an intact working memory tend to remember previously visited arms and explore less recently visited arms, whereas mice with an impaired working memory exhibit reduced spontaneous alternation behavior. In our study, LPS treatment triggered short-term spatial memory dysfunction compared to the control group. However, treatment with CX3CR1^+^ Tregs enhanced spontaneous alternation behavior, suggesting improved spatial working memory in LPS-induced neuroinflammatory mice (Figure 2C).

### 2.3. CX3CR1^+^ Tregs Home to Sites of Neuroinflammation and Reduce Activated Microglial Expression in an LPS-Induced Mouse Model

In mouse models induced by LPS, amyloid-beta (Aβ) primarily accumulates in the hippocampus, and microglia are activated around amyloid plaques [25,26]. Additionally, our previous research has shown that in Alzheimer’s disease models, adoptive Tregs migrate towards areas surrounding Aβ [21]. Based on this, we hypothesized that Tregs would be distributed around the hippocampus, and we confirmed this hypothesis. To visualize retrovirus-induced Tregs in specific areas of the mouse brain, we analyzed paramedian sagittal sections of CX3CR1^+^ Tregs that migrated to the brain using confocal laser scanning microscopy with antibodies against the DYKDDDDK tag (Flag tag) (Figure 3A,B). Subsequently, we performed an analysis of activated microglia using Iba1 antibodies. Iba-1 has been reported as a microglia-specific marker and is widely used for detecting microglia [27]. The immunofluorescence images showed increased reactivity of Iba1 in the LPS-treated mice compared to the control group. Interestingly, CX3CR1^+^ Treg treatment significantly decreased the immunofluorescence reactivity of Iba1 in the DG region of the hippocampus compared to the LPS-treated group (Figure 3C).

### 2.4. CX3CR1^+^ Tregs Effectively Suppress the Activation of Proinflammatory and Neuroinflammation-Associated Markers Induced by LPS in the Mouse Brain

In our study using the LPS-induced inflammation mouse model, we explored whether CX3CR1^+^ Tregs are more effective in suppressing microglial activation compared to CTRL-Tregs. We analyzed proinflammatory gene expressions such as CD86, iNOS, IL-1β, IL-12a, and IL-23 in the cortex and hippocampus using RT-qPCR, with GAPDH as an internal control to calculate fold changes in gene expression (Figure 4). In the cortex, apart from CD86, there were no significant differences in gene expression between the LPS and Treg treatments. However, in the hippocampus, the administration of Tregs significantly reduced the expression of these proinflammatory genes, including IL-12a, which plays a dual role in both promoting and regulating inflammatory and anti-inflammatory processes [28]. Notably, the injection of CX3CR1^+^ Tregs showed a more potent anti-inflammatory effect compared to the untreated LPS group. These data indicate that Tregs can modulate gene expression related to inflammation and that CX3CR1^+^ Tregs, especially in the hippocampus, have an enhanced ability to mitigate inflammatory responses.

Next, we investigated the effects of Treg treatments on the expression of the proinflammatory cytokines TNF-α, IL-6, and IL-1β in mice exposed to LPS, a potent activator of microglia known to trigger significant inflammatory responses in the cortex and hippocampus. Upon LPS administration, ELISA results confirmed a pronounced increase in the levels of these cytokines compared to the control, with TNF-α and IL-6 levels significantly elevated and a similar increase observed for IL-1β, indicating the strong proinflammatory impact of LPS (Figure 5B). Treatment with CX3CR1^−^Treg and CX3CR1^+^ Treg markedly altered the expression levels of TNF-α and IL-6 post-LPS stimulation, effectively reducing these cytokines and demonstrating the therapeutic potential of Treg treatments in attenuating microglial-induced inflammation. Interestingly, the expression of IL-1β did not significantly change in the Treg-treated groups following LPS stimulation, highlighting a selective efficacy of Treg treatments in modulating specific cytokine pathways, particularly in reducing TNF-α and IL-6 but not IL-1β.

We also ascertained through Western blotting that LPS administration increased the expression of NOS2 and COX2 in the cortex and hippocampus (Figure 5A). Treg adoptive transfer along with LPS significantly reduced the relative expression of these inflammatory proteins in the hippocampus.

### 2.5. CX3CR1^+^ Tregs Alleviate Cognitive Decline in 3xTg Mouse Model

The 3xTg (triple transgenic) mouse model is widely used in AD research, as it harbors three mutations associated with familial Alzheimer’s disease. This model exhibits progressive cognitive decline and neuroinflammation, making it a valuable tool for studying the chronic aspects of neurodegenerative diseases. Building on our previous findings, which demonstrated the anti-inflammatory effects of CX3CR1-transduced Tregs in an LPS-induced acute neuroinflammation model, we extended our research to assess their efficacy in a chronic neuroinflammation model using 3xTg mice. CX3CR1^−^Flag^−^ (CTRL-Tregs) and CX3CR1^+^Flag^+^ Tregs (1 × 10^5^ cells/mouse) were adoptively transferred into 5-month-old 3xTg-AD mice via tail vein injection, while wild-type (WT) and 3xTg mice received PBS as controls (Figure 6A). To evaluate the cognitive-enhancing effects of Tregs, we assessed learning and memory using the passive avoidance test (PAT) and Y-maze (Figure 6B).

CTRL-Tregs did not induce significant cognitive improvement. However, 3xTg mice receiving 1 × 10^5^ CX3CR1^+^ Tregs stayed significantly longer in the lit chamber compared to PBS-treated 3xTg mice. Furthermore, Y-maze results showed that CX3CR1^+^ Tregs, unlike CTRL-Tregs, enhanced spontaneous alternation behavior, indicating an improvement in spatial working memory function in 3xTg mice.

### 2.6. CX3CR1^+^ Tregs Home to Sites of Neuroinflammation and Reduce Activated Microglial Expression in a 3xTg Mouse Model

To visualize retrovirus-induced Tregs in the brain subregions of mice, we analyzed paramedian sagittal sections of CX3CR1^+^ Tregs that had migrated to the brain using confocal laser scanning microscopy, with antibodies against the DYKDDDDK (Flag) tag in 3xTg mice (Figure 7A). The increase in relative Flag-tag expression was most prominent in the hippocampal region.

To compare the therapeutic effects of CTRL-Tregs and CX3CR1^+^ Tregs on AD pathology, we stained the CA1 and dentate gyrus (DG) of the hippocampus with Aβ (Figure 7B). While CTRL-Tregs reduced Aβ accumulation, CX3CR1^+^ Tregs induced a more dramatic reduction.

We also analyzed activated microglia using antibodies against Iba1. The immunofluorescence images showed increased Iba1 reactivity in 3xTg mice compared to the non-transgenic control (WT) (Figure 7C). Interestingly, CX3CR1^+^ Treg treatment significantly decreased Iba1 immunoreactivity in the CA1 region of the hippocampus compared to 3xTg mice. In contrast, CTRL-Treg treatment did not result in significant suppression.

### 2.7. CX3CR1^+^ Tregs Suppress Disease-Associated Microglial Activation

To evaluate activated microglia in 3xTg mice, we analyzed the mRNA expression levels of various DAM markers in the cortex and hippocampus (Figure 8). CX3CR1 is an important receptor that regulates neuron–microglia interactions during neuroinflammation [29], and CSF1 is a growth factor necessary for microglial proliferation and activation [30]. Both of these markers showed increased expression in 3xTg mice, indicating enhanced microglial activation. CX3CR1^+^ Treg treatment suppressed this increased expression in the cortex and hippocampus, demonstrating a reduction in microglial activation. CCL6 and TREM2, which are associated with immune cell recruitment and neuroprotection, respectively [31,32,33], as well as ITGAX, a marker involved in microglial activation, did not show significant results [34]. TMEM119, a microglial marker present in a normal brain, tends to decrease in expression in mouse models with brain injury. Microglia do not simply increase inflammation; they also play a dual role by regulating inflammation through phagocytic activity [35]. CX3CR1^+^ Tregs selectively increased TMEM119 expression in the cortex, which is associated with microglial phagocytic activity. This suggests that CX3CR1^+^ Tregs may play a significant role in promoting the phagocytic function of microglia, potentially aiding in the removal of neurotoxic substances from the brain [36].

### 2.8. Engineered Tregs Suppress Activated Proinflammatory Markers

Although it has been reported that Treg attenuates proinflammatory microglial activity, it is unclear whether CX3CR1^+^ Tregs more effectively suppress the expression of activated microglia than CTRL-Tregs. To address this issue, the proinflammatory gene expressions CD86, iNOS, IL-1β, IL-12a, and IL-23 in the cortex and hippocampus were evaluated using RT-qPCR, with β-Actin used as an internal control to calculate fold changes in gene expression (Figure 9A). CTRL-Tregs did not induce the suppression of most proinflammatory gene expressions in the cortex and hippocampus, except for the expression of IL-12a in the cortex. However, CX3CR1^+^ Tregs had a significant effect on suppressing the proinflammatory mRNA expressions of the hippocampus compared to 3xTg or even to CTRL-Tregs.

Additionally, we found that increased levels of the proinflammatory cytokines TNF-α, IL-6, and IL-1β in 3xTg were significantly decreased by CX3CR1^+^ Treg treatment using ELISA (Figure 9B). We also assessed the protein levels of CD86 and iNOS in the brain using Western blot (Figure 9C). CTRL-Tregs also decreased their expression, but CX3CR1^+^ Tregs decreased it even more. These data suggest that CX3CR1^+^ Tregs had optimal inhibitory effects on inflammation.

## 3. Discussion

Our study explored the therapeutic potential of CX3CR1^+^ Tregs in mitigating neuroinflammation associated with AD across both acute and chronic models. We utilized the LPS-induced model to mimic acute neuroinflammation and the 3xTg mouse model to assess chronic neurodegenerative processes, providing broad insights into how CX3CR1^+^ Tregs modulate neuroinflammatory pathways.

We introduced a CX3CR1 retrovirus into Tregs to optimize their migration and regulate the functional activity of microglia during neuroinflammation. The production of CX3CR1 on Tregs is crucial for attracting them to activated microglial sites and inducing adaptive immune responses, which are essential for recruiting these cells to sites of inflammation within the CNS [22,23]. Flow cytometry results from our Treg trafficking study demonstrated a significant presence of CX3CR1^+^ Tregs in the brain, confirming their enhanced homing capabilities. This indicates that CX3CR1^+^ Tregs not only reach sites of inflammation but also play a vital role in mitigating the inflammatory response, which is crucial for preventing the progression of neurodegenerative processes (Figure 1D).

In brain tissues, CX3CL1 is primarily expressed in neurons, while microglia express CX3CR1, the unique receptor for CX3CL1. This interaction is pivotal in both health and disease, as neuroinflammation caused by the hyperactivity of microglia is associated with neurodegenerative diseases such as AD [12,24]. The homing of CX3CR1-transduced Tregs to the brain was confirmed through the analysis of paramedian sagittal sections of adoptively transferred CX3CR1^+^ Treg mouse brains, utilizing confocal laser scanning microscopy with antibodies against the Flag-tag. Notably, strong Flag-tag expression was detected in forebrain regions, including the cortex and hippocampus (Figure 3A,B). Activated microglial cells, revealed by anti-Iba-1 staining in the hippocampal region, were suppressed by CX3CR1^+^ Tregs (Figure 3C).

Iba1 is a critical protein in microglia and macrophages that regulates cytoskeletal reorganization, phagocytosis, and cell motility through its interaction with F-actin. While Iba1 is generally localized in the cytoplasm, it has been reported to translocate into the nucleus under certain conditions [37,38]. As shown in Figure 3C and Figure 7C, the Treg group appeared to exhibit a co-localization trend between Iba1 (red) and nuclear signals (blue, DAPI). Co-localization analysis (Appendix A) revealed a trend of increased nuclear translocation of Iba1 following Treg administration in the LPS model. These findings suggest a potential association between Treg cells and functional changes in microglia; however, the exact significance of this trend remains unclear. While nuclear Iba1 may be involved in transcriptional regulation or the expression of inflammation-related genes, further experiments and more detailed analyses are required to confirm this possibility.

This suppression coincided with a reduction in the total mRNA and protein levels of proinflammatory markers post-LPS injection (Figure 4). Furthermore, we investigated a significant suppression effect of CX3CR1^+^ Treg on inflamed microglia, supporting the idea that CX3CR1^+^ Treg regulates the activation of microglia. The downregulation of proinflammation in the microglia of LPS-induced mice was associated with reduced cytokine levels, TNF-α, IL-6, and IL-1β, demonstrating that CX3CR1^+^ Treg is a potent regulator of locally activated microglia and might be used as a potential therapeutic target to control neurodegenerative diseases and other diseases related to inflammation (Figure 5A,B).

In expanding our research to the 3xTg model, we observed that CX3CR1^+^ Tregs significantly contribute to reducing hallmark features of Alzheimer’s disease, such as Aβ accumulation and microglial activation. The 3xTg mouse model is widely used in AD research due to its three mutations associated with familial Alzheimer’s disease. This model exhibits progressive cognitive decline and neuroinflammation, making it a valuable tool for studying the chronic aspects of neurodegenerative diseases.

To evaluate the cognitive-enhancing effects of Tregs, we assessed learning and memory using the passive avoidance test (PAT) and the Y-maze. CTRL-Tregs did not induce significant cognitive improvement, whereas the 3xTg mice receiving 1 × 10^5^ CX3CR1^+^ Tregs remained in the lit compartment significantly longer than PBS-treated 3xTg mice. Furthermore, the Y-maze results showed that CX3CR1^+^ Tregs enhanced spontaneous alternation behavior, indicating an improvement in spatial working memory function compared to CTRL-Tregs (Figure 6B).

To visualize the migration of retrovirus-induced Tregs to the brain, we analyzed paramedian sagittal sections of CX3CR1^+^ Treg mice brains (day 6 post-injection) using confocal laser scanning microscopy and antibodies against the DYKDDDDK (Flag) tag. Strong Flag-Tag expression was notably detected in forebrain regions, such as the cortex and hippocampus (Figure 7A).

Moreover, we assessed the therapeutic effects of CTRL-Tregs and CX3CR1^+^ Tregs on AD pathology by staining the CA1 and DG of the hippocampus for Aβ. While CTRL-Tregs reduced Aβ accumulation, CX3CR1^+^ Tregs induced a more dramatic reduction, further corroborating their therapeutic potential (Figure 7B).

We also examined activated microglia using antibodies against Iba1. Immunofluorescence images revealed increased Iba1 reactivity in 3xTg mice compared to non-transgenic controls (WT). Notably, CX3CR1^+^ Treg treatment significantly decreased Iba1 immunoreactivity in the CA1 region of the hippocampus compared to 3xTg mice, whereas CTRL-Treg treatment did not result in significant suppression (Figure 7C).

In addition to these findings, we characterized activated microglia in 3xTg mice by assessing the mRNA expression levels of several DAM markers in the cortex and hippocampus. Both CX3CR1 and CSF1, a growth factor essential for microglial proliferation and activation, exhibited increased expression in 3xTg mice. These elevated expressions were reduced by CX3CR1^+^ Treg treatment, highlighting their regulatory effects on microglial activation. Furthermore, CX3CR1^+^ Tregs specifically increased the expression of TMEM119, a gene associated with microglial phagocytic activity in the cortex, and also modulated the expression of CCL6 and TREM2, further demonstrating their role in reducing microglial activation (Figure 8).

These findings collectively illustrate that CX3CR1^+^ Tregs not only improve cognitive functions in the 3xTg model but also play a critical role in modulating neuroinflammation and promoting a more favorable neuroinflammatory environment (Figure 9).

In this study, we aimed to leverage the unique properties of CX3CR1-expressing Tregs, going beyond the general effects of regular Tregs. These special Tregs migrate effectively to inflammatory sites through the CX3CL1/CX3CR1 axis and suppress inflammatory responses. When both control Tregs and CX3CR1^+^ Tregs were administered in a neuroinflammatory model, although some results were not statistically significant, most data showed that CX3CR1^+^ Tregs outperformed control Tregs, with the differences being particularly noticeable in the passive avoidance behavioral tests. As illustrated in the graphic abstract, when the same number of cells was administered to mice, CX3CR1^+^ Tregs migrated predominantly to CX3CL1-enriched brain regions, demonstrating localized anti-inflammatory effects focused on inflamed areas of the brain. This suggests that CX3CR1^+^ Tregs can effectively suppress neuroinflammation and enhance neuroprotection relative to the number of cells administered.

Research on the function of Tregs and the development of Treg-based cell therapies in relation to Alzheimer’s disease has been steadily advancing [39,40]. Recent studies have focused on modifying Tregs to enhance their trafficking to the brain, enabling more targeted and effective therapies [41].

Our previous studies have validated the dose-dependent therapeutic effects of Aβ-specific Tregs in the 5xFAD model, confirming that treatment efficacy can be achieved even at relatively low doses [22]. This indicates that modified Tregs possess the ability to recognize Aβ with high target specificity and selectively migrate to inflammatory sites, thereby maximizing treatment efficiency relative to the number of cells used. Therefore, administering Tregs with high target specificity in appropriate doses could prevent side effects like Cytokine Release Syndrome (CRS) [42] and precisely target inflammatory sites, potentially minimizing complications associated with systemic immunosuppression [43]. Likewise, CX3CR1^+^ Tregs could provide a more precise and effective therapeutic approach for treating inflammatory diseases compared to traditional Tregs.

However, this study has several important limitations. The specific molecular mechanisms by which CX3CR1^+^ Tregs regulate microglial activity have not yet been fully elucidated, and a clear understanding of these pathways is essential. Moreover, while this study primarily focused on short-term outcomes observed two weeks after Treg administration, neurodegenerative diseases generally progress over much longer periods. Long-term studies are crucial to evaluate the sustainability of treatment effects and the potential side effects of CX3CR1^+^ Tregs under chronic conditions. Future research should explore the detailed mechanisms through which CX3CR1^+^ Tregs exert their effects and assess the safety and efficacy of prolonged treatments. Adjustments based on cell dosage, administration intervals, and disease severity are also required to optimize treatment strategies. Lastly, translating these findings into human clinical trials based on preclinical evidence will be critical to confirm the therapeutic potential of CX3CR1^+^ Tregs in actual patient populations.

## 4. Materials and Methods

### 4.1. Animals

Male 3xTg mice harboring transgenes encoding APP (KM670/671NL), PS1 (M146V), and tau (P301L) proteins (B6;129-Psen1tm1MpmTg (APPSwe, tauP301L)1Lfa/J), non-transgenic control mice (B6129SF2/J), Thy1.1 mice (B6.PL-Thy1^a^/CyJ), and C57BL/6 mice were obtained from Jackson Laboratory (Bar Harbor, ME, USA). All animals were housed with free access to food and water and a 12 h light/dark cycle. All animal experiments were conducted according to the rules for animal care and the guiding principles for experiments using animals and were approved by the University of Kyung Hee Animal Care and Use Committee (KHUASP(SE)-21-102). All animal studies were ethically reviewed and performed in compliance with the ARRIVE guidelines.

### 4.2. Treg Transduction

Spleens from C57BL/6 mice (7–8 weeks) used to generate Tregs were homogenized and processed by mechanical disruption. The splenocytes were passed through a 40 μm cell strainer. CD4^+^CD25^+^ Tregs were isolated using magnetic-activated cell sorting (MACS) according to the manufacturer’s protocol (CD4^+^CD25^+^ Regulatory T Cell Isolation Kit, Miltenyi Biotec, Bergisch Gladbach, Germany). Tregs were cultured for 4 days. To generate retroviral vectors, one day prior to transfection, 5 × 10^5^ platA cells/well were plated into a 6-well plate in 2 mL of Dulbecco’s Modified Eagle’s Medium (DMEM) with high-glucose L-glutamine and 10% fetal bovine serum, without antibiotics. Transient transfection of the platA was performed in Optimem (Gibco, Grand Island, NY, USA) medium with 1.6% Lipofectamine 3000 (Invitrogen, Waltham, MA, USA) and 2 μg of the packaging vector pCL-Eco (Novus, Centennial, CO, USA) in combination with a plasmid encoding for m5p-CX3CR1-2A-Flag for 48 h. Retrovirus was transduced to Tregs by adding 1/3 volume filtered supernatants of the transfected platA cells, along with fresh IL-2 and a final concentration of 6 mg/mL of protamine sulphate (Sigma Aldrich; Saint Louis, MO, USA). The Tregs were spun for 2 h at 1800 rpm at 32 °C. The next day, CX3CR1^−^ Flag^−^ Tregs (CTRL-Tregs) and CX3CR1^+^ Flag^+^ Tregs (CX3CR1^+^ Tregs) were sorted using a S3e cell sorter (Bio-Rad; Hercules, CA, USA). CTRL and CX3CR1^+^ Tregs were cultured for three days in a complete medium supplemented with IL-2 (100 UI/mL).

### 4.3. Adoptive Transfer of Mouse Tregs

CX3CR1^−^Flag^−^ and CX3CR1^+^Flag^+^ Tregs (1 × 10^5^ cells/mouse) were adoptively transferred to LPS-induced mice and 5-month-old 3xTg-AD mice via tail vein injection. After 14 days, the mice were assessed with behavior tests and sacrificed.

### 4.4. Thy 1.1 Treg Trafficking

For Tregs trafficking, CX3CR1^−^Flag^−^ and CX3CR1^+^Flag^+^ Tregs were produced from the splenocytes of Thy1.1 mice. Five-month-old 3xTg-AD mice (Thy 1.2^+^) received an adoptive transfer of 1 × 10^5^ Thy 1.1^+^ Tregs. After 7 days, the mice were sacrificed, and inguinal lymph nodes, kidneys, and brain were harvested. T cells were enriched by 30–70% Percoll (Cytiva, Marlborough, MA, USA) density gradient centrifugation from the kidney and brain. Adoptively transferred Tregs were analyzed by flow cytometry.

### 4.5. Behavior Tests

To confirm the effects of CX3CR1-transduced Treg injection, Y-maze and passive avoidance tests (PATs) were performed. The PAT was performed to estimate learning and memory capacity. The apparatus consisted of a lit chamber and a dark chamber (20 cm × 20 cm × 30 cm) divided by a door. On the training day, mice were placed in a lit chamber facing away from the door. Upon the mice entering the dark chamber, the door was closed, and foot shock (0.35 mA, 2 s) was delivered. Thirty seconds after shock, the mouse was removed. The experimental animals underwent training for the passive avoidance test over two days. On the test day, the animals were placed in a lit chamber, and the latency to enter the dark chamber was recorded. No foot shocks were administered during this session; only the time taken to enter the dark compartment was measured. This approach was designed to minimize stress on the animals while accurately assessing their memory retention. Following the passive avoidance test, the Y-maze test was conducted to evaluate spatial working memory. The Y-maze consisted of three identical arms arranged at 120° angles. The animals were placed in one arm facing the wall to begin the test. They were allowed to freely explore all three arms for 5 min, and their behavior was recorded throughout the session. After completing the Y-maze test, the animals were sacrificed.

### 4.6. Immunofluorescence Staining

For immunofluorescence, mice were anesthetized with isoflurane isoflurane (Forane solution; ChoongWae Pharma, Seoul, Republic of Korea) and were transcardially perfused with PBS. The cerebral tissues were removed, incubated overnight in 4% paraformaldehyde (PFA) in 0.1 M sodium phosphate buffer (PBS), and stored in a 30% sucrose solution. After being transferred to a 30% sucrose solution, the brains were cut into 30 μm thick sections using a cryostat microtome (Leica CM 1850; Leica Microsystems, Wetzlar, Germany) and incubated at 37 °C overnight. After three 5 min washes with PBS (pH 7.4), permeabilization was performed by heating 10 mM sodium citrate buffer (pH 6.0) at 65 °C for 20 min. The brain sections were blocked for 1 h in 1% bovine serum albumin (BSA) solution and overnight at 4 °C with a 1:100 dilution of an antibody against DYKDDDDK tag (Flag-tag; Cell Signaling Technology, Danvers, MA, USA), amyloid-beta (Aβ; Biolegend, San Diego, CA, USA), and ionized calcium-binding adapter protein 1 (Iba1; Millipore Corp, Billerica, MA, USA). On the following day, after the incubation with the primary antibodies, the sections were subsequently incubated for 2 h at RT with Alexa Fluor™ 488- or Alexa Fluor™ 594-conjugated IgG secondary antibodies (Thermo Fisher Scientific, Waltham, MA, USA) at room temperature and then washed with PBS three times for 10 min per wash. Then, the sections were stained with DAPI staining solution. Finally, fluorescence images were obtained using LSM 800 confocal laser-scanning microscope (Carl Zeiss, Oberkochen, Germany). The staining intensity was measured and quantified using ImageJ software 1.53e (https://imagej.net/ij/, accessed on 17 June 2022). Staining images were split with an ImageJ plug-in to obtain a monochromatic version. The channel corresponding to the stain of interest was used for measurement.

### 4.7. RNA Extraction and RT–qPCR Assays

Total RNA was isolated from brain tissues (cortex and hippocampus) using an easy-BLUE RNA extraction kit (iNtRON Biotechnology, Seongnam, Republic of Korea, #17061), and cDNA was synthesized using Cyclescript reverse transcriptase (Bioneer, Deajeon, Republic of Korea). The samples were prepared for RT–qPCR using a SensiFAST SYBR no-Rox kit (Bioline, London, UK, USA). The cycling conditions were 95 °C for 10 s, 55 °C for 10 s, and 72 °C for 10 s, and RT–qPCR was performed using a CFX Connect System (Bio-Rad, Thane, India). The expression of target mRNAs was analyzed by the ΔΔCt method, and mouse β-actin and GAPDH were used as an endogenous control. All fold changes were calculated relative to the control group. The base sequences of the primers are shown in Table 1.

### 4.8. Western Blot Analysis

Brain tissues were lysed with PRO-PREP (iNtRON Biotechnology, Seongnam, Republic of Korea, #17061) in ice-cold conditions (4 °C) for 20 min. The protein concentration of each sample was determined using the Bradford assay. Samples (30 μg of protein) were resolved on a 10% sodium dodecyl sulfate–polyacrylamide gel and transferred to a nitrocellulose membrane, which was blocked for 30 min in 5% skim milk at room temperature. After 30 min of blocking, the membrane was incubated overnight at 4 °C with specific antibodies against CD86 (1:1000; Cell Signaling Technology, Danvers, MA, USA), iNOS (1:2000; Santa Cruz Biotechnology, Dallas, TX, USA), or β-actin (1:5000; Santa Cruz Biotechnology, Dallas, TX, USA). After washing with Tris-buffered saline containing 25 mM Tris-Cl, 150 mM NaCl, and 0.05% Tween-20, the membrane was incubated for 2 h with a horseradish peroxidase-conjugated secondary antibody (anti-rabbit or anti-mouse IgG). Immunoreactivity was visualized with a Western Blotting Detection Reagent Kit (Thermo, Waltham, MA, USA). Antibody binding was determined using a chemiluminescence detection system. Densitometric analysis was performed using ImageJ 1.53e.

### 4.9. ELISA

To measure TNF-α, IL-6, and IL-1β and in brain tissues, protein extracts were collected and analyzed by ELISA according to the manufacturer’s protocol (R&D Systems, Minneapolis, MN, USA).

### 4.10. Statistical Analysis

GraphPad Prism 10.4.0 software (GraphPad Software Inc., San Diego, CA, USA) was used for statistical analyses. A one-way analysis of variance (ANOVA) was performed, followed by Tukey’s multiple comparison tests. Two-tailed Student’s unpaired test was performed for comparing the two groups. All experiments were performed in a blinded manner and were repeated independently under identical conditions. Data were expressed as the mean ± SEM. *p* < 0.05 was considered as significant.

## Figures and Tables

**Figure 1 ijms-25-13682-f001:**
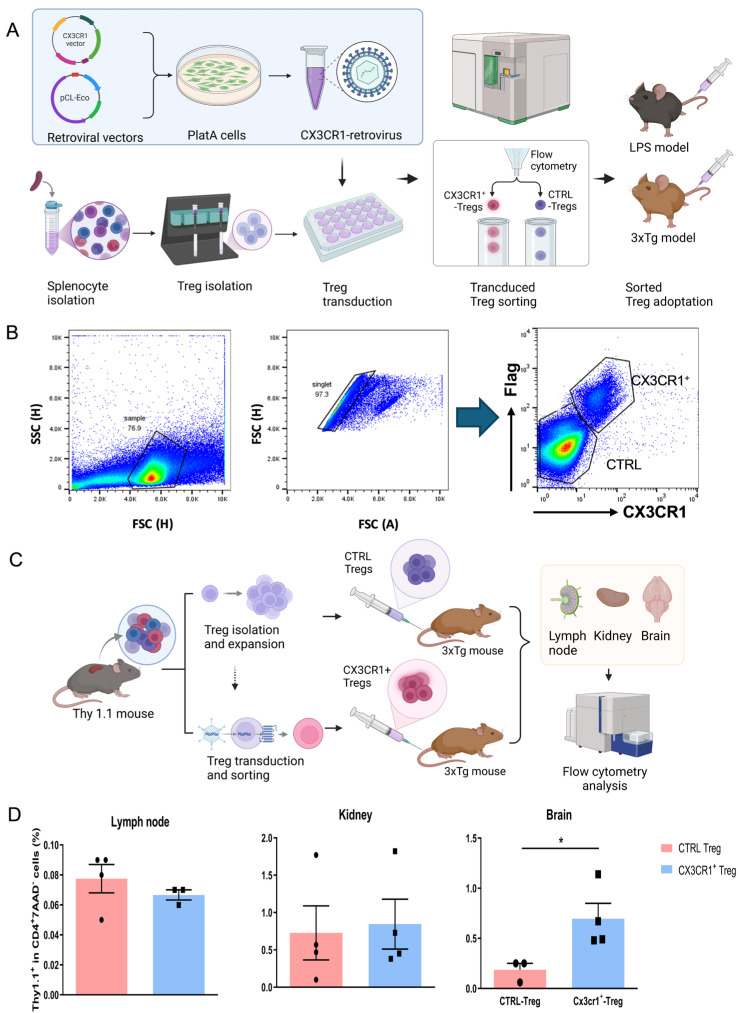
CX3CR1 retrovirus-transduced Tregs migrate to the brain. (**A**) Schematic representation of experimental workflow, engineering of CX3CR1 Treg and subsequent administration to mice. (**B**) Flow cytometry was used to identify wild-type Tregs (CX3CR1^−^Flag^−^) and transduced Tregs (CX3CR1^+^Flag^+^). Retrovirus-transduced Tregs were isolated and used for adoptive transfer. In the flow cytometry plot, colors represent cell density, with yellow indicating regions of high cell density and red representing areas of even higher cell density, reflecting a higher concentration of cells meeting the gating criteria in those regions. (**C**) The schematic depicts Thy1.1 Treg trafficking. Control and CX3CR1^+^ Tregs were generated from Thy1.1 mouse splenocytes and adoptively transferred into 3xTg-AD (Thy1.2+) mice. Seven days post-transfer, tissues (inguinal lymph nodes, kidneys, and brain) were collected, and the transferred Tregs were analyzed by flow cytometry. (**D**) Flow cytometry analysis was conducted to assess the migration of Tregs to the brain. Upon examining the lymph nodes, kidneys, and brains, a significantly higher presence of CX3CR1^+^Flag^+^ Tregs was observed in the brain tissues. Data are presented as the mean ± SEM. *n* = 5 per group; * *p* < 0.05.

**Figure 2 ijms-25-13682-f002:**
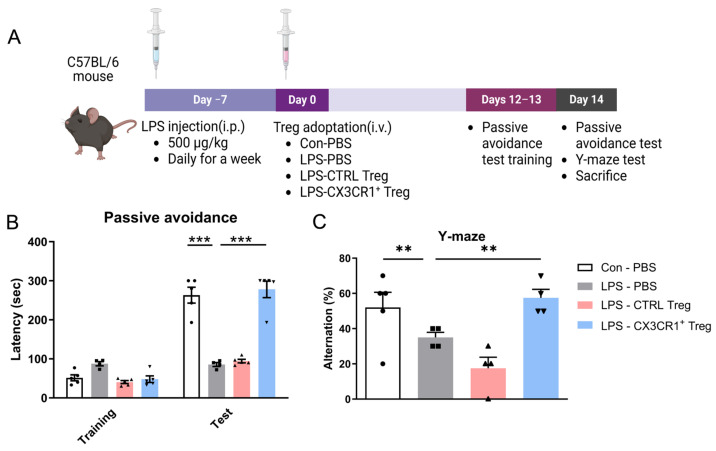
CX3CR1^+^ Treg improved the memory function of the LPS-treated mice. (**A**) Schematic representation of the workflow, including the timeline of the experiments and tests. Two weeks after the injection, mice performed the behavior test and were sacrificed for analysis of Treg homing. (**B**) In the passive avoidance test, significant intergroup differences were found in terms of the time taken to enter the dark compartment. (**C**) Spontaneous alternation behavior % of the mice during the Y-maze test. Data are presented as the mean ± SEM. *n* = 5 per group; ** *p* < 0.01, *** *p* < 0.001.

**Figure 3 ijms-25-13682-f003:**
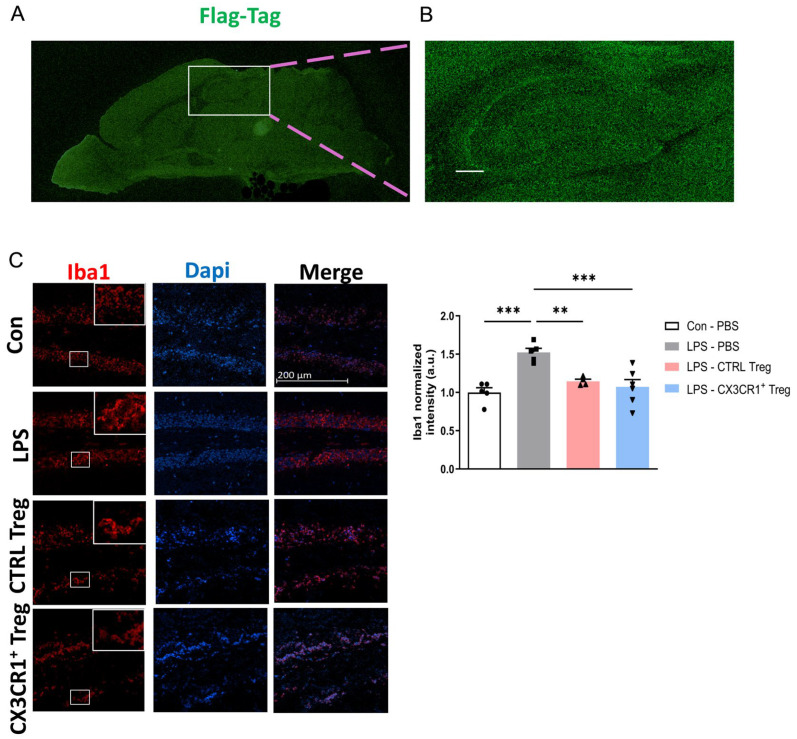
Homing of CX3CR1^+^ Treg and its effect on activated microglia expression. (**A**) Distribution of CX3CR1^+^Flag^+^ cells visualized by staining against Flag-Tag on paramedian sagittal brain section. (**B**) High-resolution analysis of CX3CR1^+^Flag^+^ cells in hippocampus region. (**C**) Representative immunofluorescence images and quantified histogram of Iba1 in the hippocampus DG region of the mice (5 mice/group). Scale bar = 200 μm. The expressed data are relative to the control. ** *p* < 0.01, *** *p* < 0.001.

**Figure 4 ijms-25-13682-f004:**
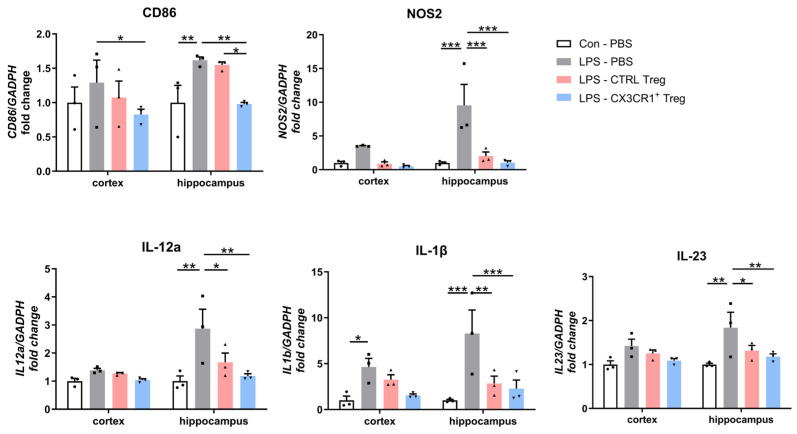
Treg downregulates proinflammatory mRNA expression. Quantitative RT-PCR analysis of genes encoding for proinflammatory markers, CD86, NOS2, IL12a, IL1b, and IL23, in cortex and hippocampus compared to control and LPS-treated group. Relative fold changes indicate mean ± SEM from three independent experiments (5 mice/group). * *p* < 0.05, ** *p* < 0.01, *** *p* < 0.001.

**Figure 5 ijms-25-13682-f005:**
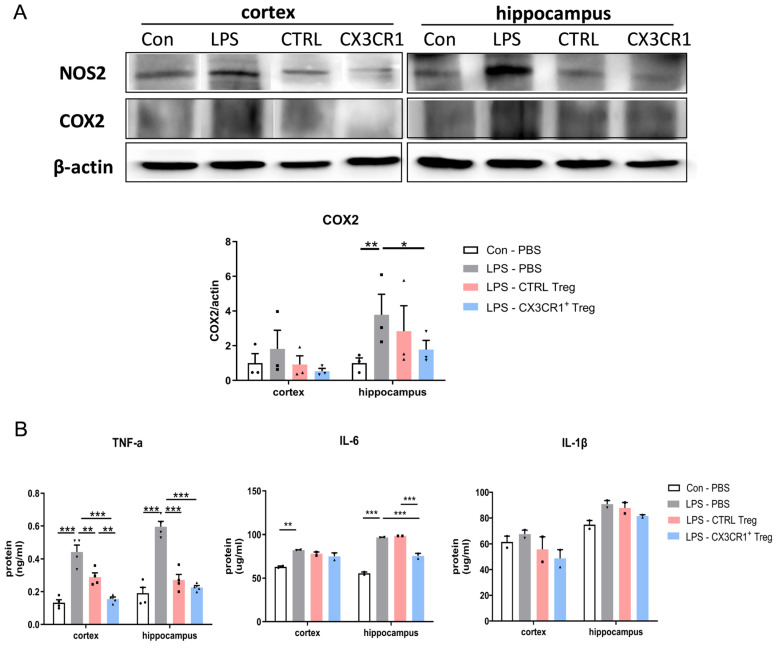
CX3CR1^+^ Tregs effectively halted the activation of inflammatory pathways and subdued the expression of proinflammatory cytokines in an LPS-induced mouse model. (**A**) Western blotting of NOS2 and COX2 proteins. The expression differences in the cortex and hippocampus of mouse brains are represented by a histogram. β-Actin was used as a loading control. (**B**) The expression of the proinflammatory cytokines TNF-α, IL-6, and IL-1β in brain homogenates. The data are described as the mean ± SEM (5 mice/group). * *p* < 0.05, ** *p* < 0.01, *** *p* < 0.001.

**Figure 6 ijms-25-13682-f006:**
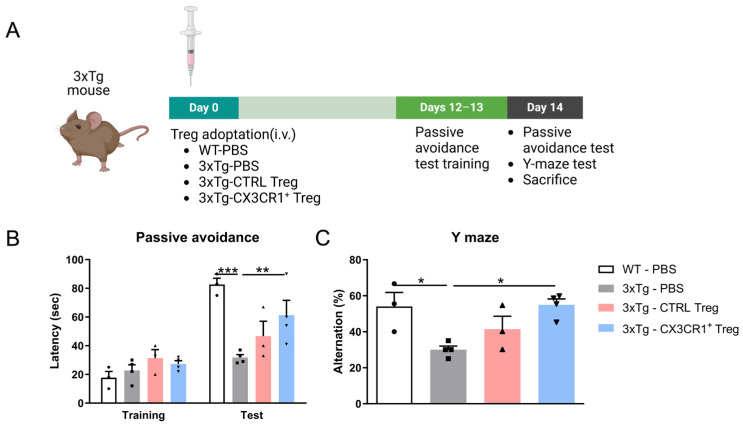
CX3CR1^+^ Tregs alleviate cognitive decline in 3xTg mouse model. (**A**) Schematic representation of the workflow, including the timeline of the experiments and tests. Two weeks after the injection, mice performed the behavior test and were sacrificed for analysis of Treg homing. (**B**) In the passive avoidance test, 3xTg mice treated with 1 × 10^5^ CX3CR1^+^ Tregs displayed significantly longer latency in entering the dark compartment compared to PBS-treated 3xTg controls, indicating improved cognitive retention. (**C**) Y-maze testing revealed that CX3CR1^+^ Tregs enhanced spontaneous alternation behavior in 3xTg mice, suggesting improved spatial working memory. Unlike control Tregs, CX3CR1^+^ Tregs showed a clear cognitive benefit in this model of Alzheimer’s disease. Data are presented as the mean ± SEM. *n* = 5 per group; * *p* < 0.05, ** *p* < 0.01, *** *p* < 0.001.

**Figure 7 ijms-25-13682-f007:**
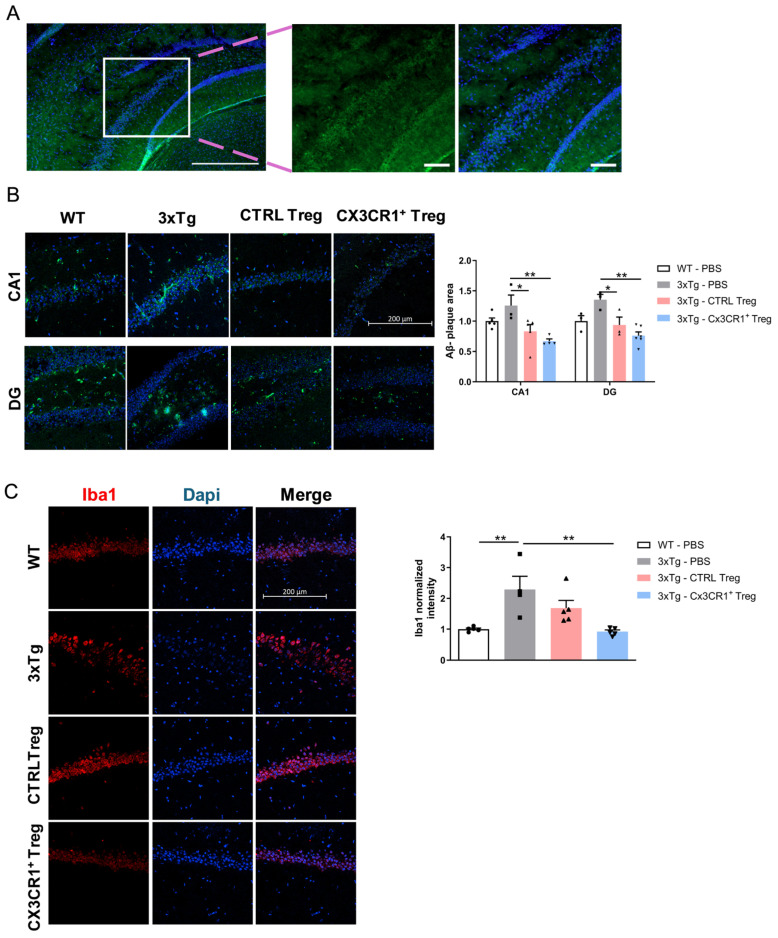
CX3CR1^+^ Tregs home to sites of neuroinflammation and reduce activated microglial expression in a 3xTg mouse model. (**A**) Visualization of CX3CR1^+^ Tregs in mouse brain subregions. Enhanced Flag-tag expression was predominantly observed in the hippocampal region, indicating targeted homing of these cells to areas of neuroinflammation. (**B**) Effect of Tregs on Aβ accumulation in the hippocampus. Immunostaining of the CA1 and dentate gyrus (DG) regions for Aβ revealed that while CTRL-Tregs led to a reduction in Aβ deposits, CX3CR1^+^ Tregs resulted in a more substantial decrease in these Alzheimer’s disease markers. (**C**) Representative immunofluorescence images and quantified histogram of Iba1 in the hippocampus DG and CA1 region of 3xTg mice. Treatment with CX3CR1^+^ Tregs significantly reduced Iba1 reactivity in the hippocampal CA1 region, surpassing the effects seen with CTRL-Tregs, which showed no significant impact on microglial activation. Scale bar = 200 μm. The expressed data are relative to the control. Data are presented as the mean ± SEM. *n* = 5 per group; * *p* < 0.05, ** *p* < 0.01.

**Figure 8 ijms-25-13682-f008:**
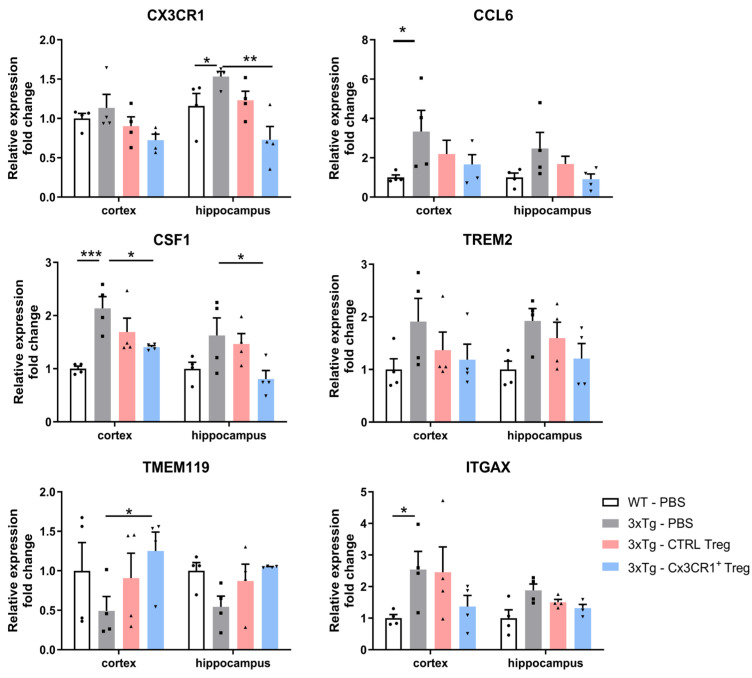
Suppression of disease-associated microglial activation by CX3CR1^+^ Tregs. Quantitative RT-PCR analysis of disease-associated microglial (DAM) markers in the cortex and hippocampus of 3xTg mice. Markers assessed include CX3CR1 and CSF1, associated with microglial activation, and TMEM119, linked to phagocytic activity. CCL6 and TREM2, related to immune cell recruitment and neuroprotection, respectively, are shown alongside ITGAX, a marker involved in microglial activation. CX3CR1 and CSF1 expressions decreased, indicating reduced activation, while TMEM119 expression increased, reflecting improved phagocytic function. Expressions of CCL6 were evaluated to demonstrate the comprehensive impact of CX3CR1^+^ Tregs on neuroinflammation. Data are presented as the mean ± SEM. *n* = 5 per group; * *p* < 0.05, ** *p* < 0.01, *** *p* < 0.001.

**Figure 9 ijms-25-13682-f009:**
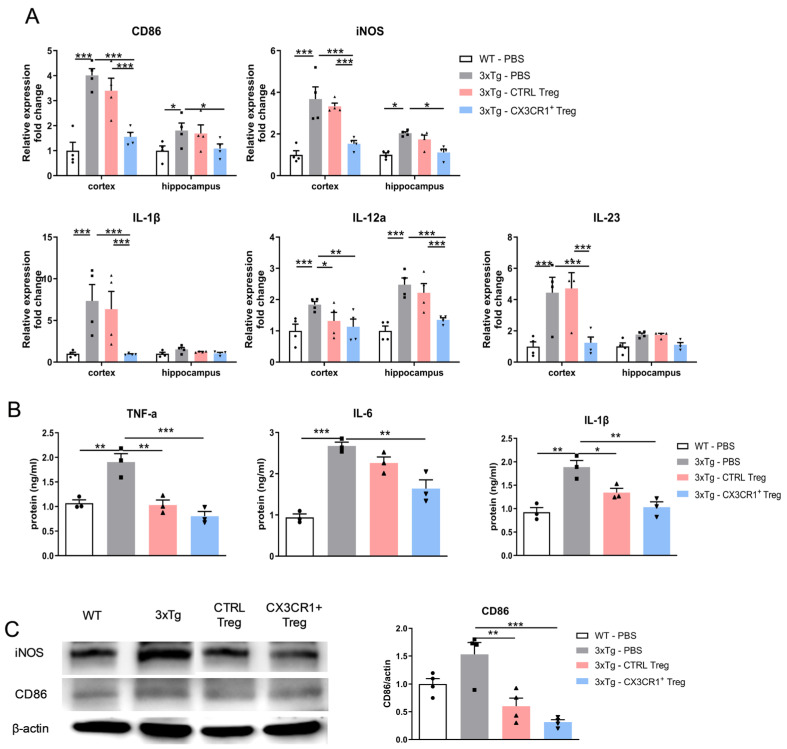
Effectiveness of CX3CR1^+^ Tregs in suppressing inflammatory responses. (**A**) RT-qPCR analysis of proinflammatory gene expression in the cortex and hippocampus. RT-qPCR was used to evaluate the expression of proinflammatory genes CD86, iNOS, IL-1β, IL-12a, and IL-23 in the cortex and hippocampus. β-Actin served as an internal control for calculating fold changes in gene expression. (**B**) ELISA results of proinflammatory cytokines in 3xTg mice. CX3CR1^+^ Tregs significantly decreased the levels of TNF-α, IL-6, and IL-1β. (**C**) Western blot analysis of CD86 and iNOS protein levels in the brain. While CTRL-Tregs reduced their expression, CX3CR1^+^ Tregs achieved a greater decrease. Data are presented as the mean ± SEM. *n* = 5 per group; * *p* < 0.05, ** *p* < 0.01, *** *p* < 0.001.

**Table 1 ijms-25-13682-t001:** The base sequences of the primers.

Gene Name	Forward Primer Sequence (5′-3′)	Reverse Primer Sequence (5′-3′)
β-Actin	GTG CTA TGT TGC TCT AGA CTT CG	ATG CCA CAG GAT TCC ATA CC
CX3CR1	GGA GAC TGG AGC CAA CAG AG	TCT TGT CTG GCT GTG TCC TG
CD86	GAC CGT TGT GTG TGT TCT GG	GAT GAG CAG CAT CAC AAG GA
iNOS	CAG CTG GGC TGT ACA AAC CTT	CAT TGG AAG TGA AGC GTT TCG
IL-1β	AAG CCT CGT GCT GTC GGA CC	TGA GGC CCA AGG CCA CAG G
IL-12a	AAG CTC TGC ATC CTG CTT CAC	GAT AGC CCA TCA CCC TGT TGA
IL-23	CCT TCT CCG TTC CAA GAT CCT	ACT AAG GGC TCA GTC AGA GTT GCT
CCL6	ATG ATG AGA CAT TCC AAG ACT GC	TCA AGC AAT GGC ACT GTT CCC AGA
ITGAX	CTG GAT AGC CTT TCT TCT GCT G	GCA CAC TGT GTC CGA ACT CA
CSF1	AGT ATT GCC AAG GAG GTG TCA G	ATC TGG CAT GAA GTC TCC ATT T
TREM2	AGG GCC CAT GCC AGC GTG TGG T	CCA GAG ATC TCC AGC ATC
TMEM119	GTG TCT AAC AGG CCC CAG AA	AGC CAC GTG GTA TCA AGG AG
GAPDH	CAA CTC CCA CTC TTC CAC CT	GAG TTG GGA TAG GGC CTC TC

## Data Availability

Data are contained within the article and Appendix A.

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
