# Peer review of "Adoptive Transfer of CX3CR1-Transduced Tregs Homing to the Forebrain in Lipopolysaccharide-Induced Neuroinflammation and 3xTg Alzheimer’s Disease Models"

_ijms, 2024, doi:10.3390/ijms252413682_

Round 1
Reviewer 1 Report (Previous Reviewer 2)
Comments and Suggestions for Authors
The authors have addressed the reviewer’s comments appropriately. Below, I suggest several additional points to improve the manuscript and enhance its acceptability:
1) The microglial marker Iba1 is predominantly localized in the cytoplasm; however, under certain conditions, it has been observed to translocate to the nucleus [1,2]. In the immunofluorescent images presented in this manuscript, both in vitro (Fig. 3) and in vivo (Fig. 7), the Iba1 signals appear to co-localize with the nucleus, particularly in the CX3CR1+Treg group. I recommend performing co-localization analyses of the blue (nuclear) and red (Iba1) signals and discussing the findings. Although the role of nuclear Iba1 remains unclear [1,2], such an analysis could provide intriguing insights and benefit potential readers in future studies.
[1] Ohsawa, Keiko, et al. "Involvement of Iba1 in membrane ruffling and phagocytosis of macrophages/microglia." Journal of Cell Science 113.17 (2000): 3073-3084.
[2] Korzhevskii, D. E., et al. "Intranuclear accumulation of Iba-1 protein in microgliocytes in the human brain." Neuroscience and Behavioral Physiology 47 (2017): 435-437.
2) While the authors have provided explanations for the genes shown in Figure 8, relevant references supporting the role of these genes should be included in the next revision to strengthen the manuscript.
3) Following recent trends in data visualization, it is recommended to display individual data points as dots on the bar charts of immunoblot results.
Author Response
Dear Reviewer,
We sincerely appreciate the time and effort the reviewers have dedicated to providing valuable comments and constructive feedback on our manuscript. Their insights have been instrumental in improving the quality of our study.
We have carefully addressed all comments and suggestions, and the revisions have been incorporated into the revised manuscript. All changes have been highlighted in the manuscript to ensure they are easily identifiable.
Below, we provide a point-by-point response to each of the reviewers’ comments, detailing the changes made to the manuscript.
Reviewer 1
The authors have addressed the reviewer’s comments appropriately. Below, I suggest several additional points to improve the manuscript and enhance its acceptability:
1) The microglial marker Iba1 is predominantly localized in the cytoplasm; however, under certain conditions, it has been observed to translocate to the nucleus [1,2]. In the immunofluorescent images presented in this manuscript, both in vitro (Fig. 3) and in vivo (Fig. 7), the Iba1 signals appear to co-localize with the nucleus, particularly in the CX3CR1+Treg group. I recommend performing co-localization analyses of the blue (nuclear) and red (Iba1) signals and discussing the findings. Although the role of nuclear Iba1 remains unclear [1,2], such an analysis could provide intriguing insights and benefit potential readers in future studies.
[1] Ohsawa, Keiko, et al. "Involvement of Iba1 in membrane ruffling and phagocytosis of macrophages/microglia." Journal of Cell Science 113.17 (2000): 3073-3084.
[2] Korzhevskii, D. E., et al. "Intranuclear accumulation of Iba-1 protein in microgliocytes in the human brain." Neuroscience and Behavioral Physiology 47 (2017): 435-437.
We thank you for your kind review and insightful suggestions regarding the interpretation of Iba1 staining. Your comments, along with the appropriate references, enabled us to conduct a more comprehensive analysis. We performed co-localization analysis of Iba1 (red) and nuclear signals (blue) using the ImageJ/FIJI program. The results of this analysis have been included as a supplementary figure for further clarification, and we have addressed this in the discussion section.
This revision has been incorporated into the manuscript as follows:
340-351> Iba1 is a critical protein in microglia and macrophages that regulates cytoskeletal reorganization, phagocytosis, and cell motility through its interaction with F-actin. While Iba1 is generally localized in the cytoplasm, it has been reported to translocate into the nucleus under certain conditions[37, 38]. In Figures 3C and 7C, the Treg group appeared to exhibit a co-localization trend between Iba1 (red) and nuclear signals (blue, DAPI). Co-localization analysis (Supplementary Figure 1) revealed a trend of increased nuclear translocation of Iba1 following Treg administration in the LPS model. These findings suggest a potential association between Treg cells and functional changes in microglia; however, the exact significance of this trend remains unclear. While nuclear Iba1 may be involved in transcriptional regulation or the expression of inflamma-tion-related genes, further experiments and more detailed analyses are required to confirm this possibility.
2) While the authors have provided explanations for the genes shown in Figure 8, relevant references supporting the role of these genes should be included in the next revision to strengthen the manuscript.
Thank you for pointing this out. We recognize that relevant references supporting the roles of the genes shown in Figure 8 were missing. To address this, we have reviewed key studies related to the functions of CX3CR1, CSF1, CCL6, TREM2, ITGAX.. In the revised version, we will incorporate the following references to strengthen the explanations provided in the manuscript.
This revision has been incorporated into the manuscript as follows:
261-269> CX3CR1 is an important receptor that regulates neuron-microglia interactions during neuroinflammation[1], and CSF1 is a growth factor necessary for microglial proliferation and activation [2]. Both of these markers showed increased expression in 3xTg mice, indicating enhanced microglial activation. CX3CR1+ Treg treatment suppressed this increased expression in the cortex and hippocampus, demonstrating a reduction in microglial activation. CCL6 and TREM2, which are associated with immune cell recruitment and neuroprotection respectively [3-5], as well as ITGAX, a marker involved in microglial activation, did not show significant results [6].
3) Following recent trends in data visualization, it is recommended to display individual data points as dots on the bar charts of immunoblot results.
Thank you for your suggestion regarding data visualization. We agree that displaying individual data points as dots on the bar charts for the immunoblot results would improve transparency and provide a clearer representation of the data distribution. In the revised manuscript, we will modify the bar charts to include individual data points, ensuring alignment with current trends in data presentation and enhancing the clarity of the results
- Pawelec, P.; Ziemka-Nalecz, M.; Sypecka, J.; Zalewska, T., The Impact of the CX3CL1/CX3CR1 Axis in Neurological Disorders. Cells 2020, 9, (10).
- Elmore, M. R.; Najafi, A. R.; Koike, M. A.; Dagher, N. N.; Spangenberg, E. E.; Rice, R. A.; Kitazawa, M.; Matusow, B.; Nguyen, H.; West, B. L.; Green, K. N., Colony-stimulating factor 1 receptor signaling is necessary for microglia viability, unmasking a microglia progenitor cell in the adult brain. Neuron 2014, 82, (2), 380-97.
- Zhong, L.; Xu, Y.; Zhuo, R.; Wang, T.; Wang, K.; Huang, R.; Wang, D.; Gao, Y.; Zhu, Y.; Sheng, X.; Chen, K.; Wang, N.; Zhu, L.; Can, D.; Marten, Y.; Shinohara, M.; Liu, C.-C.; Du, D.; Sun, H.; Wen, L.; Xu, H.; Bu, G.; Chen, X.-F., Soluble TREM2 ameliorates pathological phenotypes by modulating microglial functions in an Alzheimer’s disease model. Nature Communications 2019, 10, (1), 1365.
- Kanno, M.; Suzuki, S.; Fujiwara, T.; Yokoyama, A.; Sakamoto, A.; Takahashi, H.; Imai, Y.; Tanaka, J., Functional expression of CCL6 by rat microglia: A possible role of CCL6 in cell–cell communication. Journal of Neuroimmunology 2005, 167, (1), 72-80.
- Deczkowska, A.; Keren-Shaul, H.; Weiner, A.; Colonna, M.; Schwartz, M.; Amit, I., Disease-Associated Microglia: A Universal Immune Sensor of Neurodegeneration. Cell 2018, 173, (5), 1073-1081.
- Wlodarczyk, A.; Holtman, I. R.; Krueger, M.; Yogev, N.; Bruttger, J.; Khorooshi, R.; Benmamar‐Badel, A.; de Boer‐Bergsma, J. J.; Martin, N. A.; Karram, K.; Kramer, I.; Boddeke, E. W. G. M.; Waisman, A.; Eggen, B. J. L.; Owens, T., A novel microglial subset plays a key role in myelinogenesis in developing brain. The EMBO Journal 2017, 36, (22), 3292-3308-3308.

Reviewer 2 Report (New Reviewer)
Comments and Suggestions for Authors
Thank you for the opportunity to review this highly interesting manuscript. It presents a significant contribution to the field, offering robust scientific insights and well-structured experimentation. The study is well organized, with clearly defined objectives that are successfully addressed using compelling evidence.
The research focuses on the therapeutic potential of CX3CR1+ regulatory T cells in modifying neuroinflammation associated with Alzheimer’s disease across both acute (LPS-induced neuroinflammation model) and chronic (3xTg-AD mouse model) settings. Specifically, the study examines the adoptive transfer of CX3CR1-Flag- and CX3CR1+Flag+ Tregs into these models. The successful migration of transferred cells into brain tissue is experimentally validated, strengthening the study’s mechanistic insights. Behavioral assessments, including the Y-maze test and passive avoidance test, are used to evaluate cognitive functions, while markers of neuroinflammation, microglial activation, and disease-associated microglial phenotypes are systematically analyzed. The manuscript is methodically structured, with clear alignment between its objectives, methods, results, and discussion. The authors also address major limitations, enhancing the credibility of their findings.
I have a few suggestions for improving the manuscript:
1. Irrelevant significance values in the figure legends should be removed, particularly in Figures 1 (** p < 0.01, *** p < 0.001), Figure 2 (** p < 0.01), and Figure 7 (*** p < 0.001).
2. The timeline of the behavioral experiments requires clarification. From Day 11 to Day 13, the training sessions for the passive avoidance test were conducted, followed by the Y-maze test on Day 13 and the passive avoidance test on Day 14. What is the rationale behind this mixed procedure? Furthermore, during the three-day training session for the passive avoidance test, the mice were subjected to foot shocks. Were these procedures likely to induce stress in the animals? If so, what precautions were taken to minimize stress and ensure the animals’ well-being?
3. In Figure 2, the Y-maze results appear to be missing significance levels. If there are no significant changes, how was the conclusion about the enhancement of spontaneous behavior with CX3CR1+ treatment reached (refer to lines 128–131)? Clarification is needed to justify this interpretation or revise the conclusion accordingly.
4. Refer to lines 217–219: “3xTg mice receiving 1 x 10⁵ CX3CR1+ Tregs remained in the dark compartment significantly longer than PBS-treated 3xTg mice” requires clarification. The results in Figure 6B describe latency (seconds), which represents the time taken to enter the dark compartment. How does this align with the statement above? The explanation seems contradictory and needs to be addressed. Additionally, similar attention is required in the discussion section (lines 357–358) to ensure consistency in the interpretation of the results.
5. The discussion section is well explained; however, it primarily focuses on the present results. The interpretation and conclusions would be strengthened by integrating more related evidence from previous research. Currently, only three supportive studies are cited, which limits the broader context of the findings. Incorporating additional relevant studies from earlier research would provide a more comprehensive and robust foundation for the discussion.
Author Response
Dear Reviewer,
We sincerely appreciate the time and effort the reviewers have dedicated to providing valuable comments and constructive feedback on our manuscript. Their insights have been instrumental in improving the quality of our study.
We have carefully addressed all comments and suggestions, and the revisions have been incorporated into the revised manuscript. All changes have been highlighted in the manuscript to ensure they are easily identifiable.
Below, we provide a point-by-point response to each of the reviewers’ comments, detailing the changes made to the manuscript.
Reviewer 2
Thank you for the opportunity to review this highly interesting manuscript. It presents a significant contribution to the field, offering robust scientific insights and well-structured experimentation. The study is well organized, with clearly defined objectives that are successfully addressed using compelling evidence.
The research focuses on the therapeutic potential of CX3CR1+ regulatory T cells in modifying neuroinflammation associated with Alzheimer’s disease across both acute (LPS-induced neuroinflammation model) and chronic (3xTg-AD mouse model) settings. Specifically, the study examines the adoptive transfer of CX3CR1-Flag- and CX3CR1+Flag+ Tregs into these models. The successful migration of transferred cells into brain tissue is experimentally validated, strengthening the study’s mechanistic insights. Behavioral assessments, including the Y-maze test and passive avoidance test, are used to evaluate cognitive functions, while markers of neuroinflammation, microglial activation, and disease-associated microglial phenotypes are systematically analyzed. The manuscript is methodically structured, with clear alignment between its objectives, methods, results, and discussion. The authors also address major limitations, enhancing the credibility of their findings.
I have a few suggestions for improving the manuscript:
1.Irrelevant significance values in the figure legends should be removed, particularly in Figures 1 (** p < 0.01, *** p < 0.001), Figure 2 (** p < 0.01), and Figure 7 (*** p < 0.001).
Thank you for bringing this to our attention. We fully agree that including irrelevant significance values in the figure legends could cause confusion. In response, we have thoroughly reviewed and revised all Figures to remove any unnecessary significance indicators. Additionally, we have carefully double-checked these revisions to ensure the accuracy and clarity of the manuscript. We sincerely appreciate your insightful feedback, which has greatly contributed to improving the presentation of our data.
- The timeline of the behavioral experiments requires clarification. From Day 11 to Day 13, the training sessions for the passive avoidance test were conducted, followed by the Y-maze test on Day 13 and the passive avoidance test on Day 14. What is the rationale behind this mixed procedure? Furthermore, during the three-day training session for the passive avoidance test, the mice were subjected to foot shocks. Were these procedures likely to induce stress in the animals? If so, what precautions were taken to minimize stress and ensure the animals’ well-being?
First, we would like to apologize for the slight error in describing the behavioral experiment timeline, which may have caused some confusion. Thank you for pointing this out. The training sessions for the passive avoidance test were conducted over two days, and on the third day, only the latency to enter the dark compartment was recorded without administering any foot shocks. Later on the same day, the Y-maze test was performed, followed by the sacrifice of the animals. We have revised the figure to accurately reflect this timeline.
Regarding the foot shocks used during the passive avoidance test, the shock intensity was minimized to 0.35 mA, and the duration was limited to 2 seconds. These settings were carefully selected to ensure effective training while minimizing potential stress to the animals. During and after the training sessions, the animals were closely observed for any signs of abnormal behavior or stress. Adequate recovery periods were provided between trials to allow the animals to regain a calm state, and the experimental environment was maintained in a quiet and stable condition to minimize external stressors.
Additionally, all procedures were conducted in strict compliance with institutional and international ethical guidelines for animal research, and the experimental protocols were reviewed and approved by the institutional ethics committee.
This revision has been incorporated into the manuscript as follows:
488-498> The experimental animals underwent training for the passive avoidance test over two days. On the test day, the animals were placed in a lighted chamber, and the latency to enter the dark chamber was recorded. No foot shocks were administered during this session; only the time taken to enter the dark compartment was measured. This approach was designed to minimize stress on the animals while accurately assessing their memory retention. Following the passive avoidance test, the Y-maze test was conducted to evaluate spatial working memory. The Y-maze consisted of three identical arms arranged at 120° angles. The animals were placed in one arm facing the wall to begin the test. They were allowed to freely explore all three arms for 5 minutes, and their behavior was recorded throughout the session. After completing the Y-maze test, the animals were sacrificed.
- In Figure 2, the Y-maze results appear to be missing significance levels. If there are no significant changes, how was the conclusion about the enhancement of spontaneous behavior with CX3CR1+ treatment reached (refer to lines 128–131)? Clarification is needed to justify this interpretation or revise the conclusion accordingly.
Thank you for pointing out the issue with the Y-maze results in Figure 2. Upon review, we discovered that the significance levels were inadvertently omitted in the figure. We have carefully re-examined all graphs in the manuscript and made the necessary revisions to ensure consistency and accuracy in the presentation of our data. Specifically, the Y-maze results demonstrate a significant improvement in spontaneous alternation behavior with CX3CR1+ Treg treatment. We have added the missing significance levels and revised the conclusion to accurately reflect the data. Your insightful feedback has been invaluable in enhancing the clarity and accuracy of our manuscript. Thank you once again for your thoughtful suggestions.
This revision has been incorporated into the manuscript as follows:
121-129> The Y-maze test is used to evaluate short-term spatial working memory in mice. Spontaneous alternation behavior, a measure of spatial working memory, is driven by the innate curiosity of rodents to explore new environments. Mice with intact working memory tend to remember previously visited arms and explore less recently visited arms, whereas mice with impaired working memory exhibit reduced spontaneous al-ternation behavior. In our study, LPS treatment triggered short-term spatial memory dysfunction compared to the control group. However, treatment with CX3CR1+ Tregs enhanced spontaneous alternation behavior, suggesting improved spatial working memory in LPS-induced neuroinflammatory mice (Figure 2C).
- Refer to lines 217–219: “3xTg mice receiving 1 x 10⁵ CX3CR1+ Tregs remained in the dark compartment significantly longer than PBS-treated 3xTg mice” requires clarification. The results in Figure 6B describe latency (seconds), which represents the time taken to enter the dark compartment. How does this align with the statement above? The explanation seems contradictory and needs to be addressed. Additionally, similar attention is required in the discussion section (lines 357–358) to ensure consistency in the interpretation of the results.
Thank you for your valuable feedback. The results indicate that the mice stayed significantly longer in the lit chamber, not the dark chamber. We acknowledge this mistake in the wording and sincerely apologize for any confusion it may have caused. We are grateful for your observation, which allowed us to correct this error.
Additionally, we have reviewed and corrected the discussion section to ensure consistency with this interpretation.
We sincerely appreciate your detailed feedback, which helped us improve the clarity and accuracy of our manuscript. Thank you once again.
This revision has been incorporated into the manuscript as follows:
215-217> 3xTg mice receiving 1 x 10⁵ CX3CR1+ Tregs stayed significantly longer in the lighted chamber compared to PBS-treated 3xTg mice.
368-371> CTRL-Tregs did not induce significant cognitive improvement, whereas the 3xTg mice receiving 1 x 105 CX3CR1+ Tregs remained in the lighted compartment significantly longer than PBS-treated 3xTg mice.
- The discussion section is well explained; however, it primarily focuses on the present results. The interpretation and conclusions would be strengthened by integrating more related evidence from previous research. Currently, only three supportive studies are cited, which limits the broader context of the findings. Incorporating additional relevant studies from earlier research would provide a more comprehensive and robust foundation for the discussion.
Thank you for your thoughtful feedback and valuable suggestions. We appreciate your recommendation to incorporate more evidence from previous research to strengthen the interpretation and conclusions in the discussion section. In response, we have added additional references to relevant studies, particularly focusing on earlier research that aligns with our findings. These additions aim to broaden the foundation of our discussion and offer a more comprehensive perspective on the implications of our results. Thank you again for your insightful comments, which have greatly helped improve the quality of our manuscript.
This revision has been incorporated into the manuscript as follows:
412-415> Research on the function of Tregs and the development of Treg-based cell therapies in relation to Alzheimer’s disease has been steadily advancing [39, 40]. Recent studies have focused on modifying Tregs to enhance their trafficking to the brain, enabling more targeted and effective therapies [41].
- Dansokho, C.; Ait Ahmed, D.; Aid, S.; Toly-Ndour, C.; Chaigneau, T.; Calle, V.; Cagnard, N.; Holzenberger, M.; Piaggio, E.; Aucouturier, P.; Dorothée, G., Regulatory T cells delay disease progression in Alzheimer-like pathology. Brain 2016, 139, (4), 1237-1251.
- Faridar, A.; Thome, A. D.; Zhao, W.; Thonhoff, J. R.; Beers, D. R.; Pascual, B.; Masdeu, J. C.; Appel, S. H., Restoring regulatory T-cell dysfunction in Alzheimer's disease through ex vivo expansion. Brain Commun 2020, 2, (2), fcaa112.
- Yeapuri, P.; Machhi, J.; Lu, Y.; Abdelmoaty, M. M.; Kadry, R.; Patel, M.; Bhattarai, S.; Lu, E.; Namminga, K. L.; Olson, K. E.; Foster, E. G.; Mosley, R. L.; Gendelman, H. E., Amyloid-β specific regulatory T cells attenuate Alzheimer's disease pathobiology in APP/PS1 mice. Mol Neurodegener 2023, 18, (1), 97.

Round 2
Reviewer 1 Report (Previous Reviewer 2)
Comments and Suggestions for Authors
The authors have addressed the reviewer’s comments appropriately. The manuscript is so interesting that it will attract potential readers in IJMS.
This manuscript is a resubmission of an earlier submission. The following is a list of the peer review reports and author responses from that submission.
Round 1
Reviewer 1 Report
Comments and Suggestions for Authors
The manuscript submitted by Hyunsu Bae hypothesizes the potential role of CX3CR1-Transduced Tregs in reducing neuroinflammation by targeting microglial activation. Even though it is a hot topic gaining interest of the scientific community, the experimental data presented are not enough to confirm the authors’ hypothesis. In most of the experiments proposed, the neuroinflammation was counteracted even by non-transduced Treg making the conclusion too speculative and not supported by strong data.
The manuscript is not suitable for publication since the experiments proposed do not verify the initial hypothesis.
Find attached a PDF file with more detailed comments.

Author Response
Thank you for your valuable and insightful reviews. We deeply appreciate the opportunity to respond to your comments and incorporate your suggestions, which have allowed us to further improve the quality and precision of our manuscript.
We have provided detailed responses to each of your questions, with answers formatted in red and italic for clarity, and the revisions are clearly indicated with underlines in the response letter. Changes within the manuscript are highlighted for ease of review. Additionally, to facilitate a better understanding of the experimental content, we have significantly revised the graphic abstract.
Comments 1: The first assumption is related to the expression of Flag-tag Lane 145 “Since the increase in the relative expression of Flag-tag was most prominent in the hippocampus region”. How do you say that? Can you please show the expression of Flag-tag in different brain regions and the respect quantification to prove the most prominent expression in the hippocampus region
We appreciate your attention to the distribution of Flag-tagged Tregs mentioned in our paper. After reviewing your comments and re-evaluating the data presented in Figure 1C, we concur with your observations. Our data confirmed that CX3CR1+ Flag+ Tregs migrate to the brain. However, without specific quantification, the assertion that these Tregs predominantly increase in the hippocampus region was not adequately supported. Previous research shows that in LPS-induced mouse brain models, amyloid-beta (Aβ) primarily accumulates in the hippocampus, where microglia activate and cluster around damaged neurons and amyloid plaques (1, 2). Furthermore, our previous studies in Alzheimer's disease models demonstrated that adoptive Tregs migrate towards areas surrounding Aβ (3). Building on these findings, we hypothesized that Flag-tagged Tregs are abundantly distributed around the hippocampus. We confirmed this hypothesis through confocal imaging and further analyzed the activated microglia using Iba1 antibodies (Fig. 3C).
The corrected sentence will read:
(148-158) In mouse models induced by LPS, amyloid-beta (Aβ) primarily accumulates in the hippocampus, and microglia are activated around amyloid plaques (1, 2). Additionally, our previous research has shown that in Alzheimer's disease models, adoptive Tregs migrate towards areas surrounding Aβ (3). Based on this, we hypothesized that Tregs would be distributed around the hippocampus, and we confirmed this hypothesis. To visualize retrovirus-induced Tregs in specific areas of the mouse brain, we analyzed paramedian sagittal sections of CX3CR1+ Tregs that migrated to the brain using confocal laser scanning microscopy with antibodies against the DYKDDDDK tag (Flag tag) (Fig. 3A, B). Subsequently, we performed an analysis of activated microglia using Iba1 antibodies. Subsequently, we performed an analysis of activated microglia using Iba1 antibodies. Iba-1 has been reported as a microglia-specific marker and is widely used for detecting microglia (4).
Comments 2: 3C. Please improve the quality of immunofluorescence representative figure and provide higher magnification images.
Thank you for your valuable feedback regarding Fig. 3C. Unfortunately, we were unable to capture higher magnification images at this time. Instead, we will highlight and magnify key areas of interest within the existing images and enhance the overall DPI to improve clarity and better convey our results.
Comments 3: 4A. Nos2 expression decreased in both CX3CR1- and CXCR1+ vs LPS indicating that the reduction of inflammation is triggered by Treg even without expression of CXCR1. This is also the case in other results, and it contradicts your hypothesis about the role of CX3CR1- Transduced Tregs. Can you comment on this result? Same consideration for IL-1B, IL-12a and IL-23. Also, IL-12a acts as an anti-inflammatory molecule and in your experiment, it is upregulated in response to LPS treatment. I would say that the result in fig 4 (paragraph 2.4) are not supported by experimental data. The effect on inflammatory marker is not specific for CXCR1+. Therefore, the initial hypothesis was not tested experimentally.
I would like to thank you for your insightful comments. It seems there was some misunderstanding regarding the interpretation of our research results, which I would like to clarify. Previous studies have demonstrated that the administration of Tregs can have neuroprotective effects in Alzheimer’s disease (AD) (refer to Introduction, lines 63-67). In this study, we aimed to explore beyond the general effects of Tregs, utilizing the properties of CX3CR1-expressing Tregs which effectively migrate to inflammatory sites via the CX3CL1/CX3CR1 axis to suppress inflammatory responses. While general Tregs can mitigate neuroinflammation induced by LPS, the injection of cx3cr1+ Tregs resulted in a greater number of cells migrating to the brain, thus observing a more potent anti-inflammatory effect.
IL-12 plays a critical dual role in the immune system, both enhancing and regulating immune responses. This cytokine, a heterodimer composed of IL-12a (p35) and IL-12b (p40) subunits, is primarily produced by activated macrophages and dendritic cells. IL-12 boosts the immune response against pathogens by promoting the differentiation of naive T cells into Th1 cells and enhancing the production of IFN-γ, which increases the pathogen-clearing capabilities of macrophages and amplifies inflammatory responses. However, IL-12 also contributes to anti-inflammatory processes, helping to maintain immune homeostasis and regulate disease. Thus, IL-12 exemplifies the complex interplay between pro-inflammatory and anti-inflammatory roles, crucial for balancing immune activation and suppression in both health and disease (5).
The corrected sentence will read:
(176-184) In the cortex, apart from CD86, there are no significant differences in gene expression between the LPS and Treg treatments. However, in the hippocampus, the administration of Tregs significantly reduces the expression of these genes, including IL-12a, which plays a dual role in both promoting and regulating inflammatory and anti-inflammatory processes (5). Notably, the injection of CX3CR1+ Tregs shows a more potent anti-inflammatory effect compared to the untreated LPS group. This data indicates that Tregs can modulate gene expression related to inflammation and that CX3CR1+ Tregs, especially in the hippocampus, have an enhanced ability to mitigate inflammatory responses.
Comments 4: 5A. As above. Both CX3CR1- and CX3CR1+ reduced the upregulation of NOS2 expression due to LPS treatment. The conclusions are not supported by experimental data. Also, the western blot representative figure for COX2 has high background and no bands. I would suggest removing that figure and quantification from the manuscript if you do not get a higher quality result. In my experience, it is impossible to analyze and make a quantification of protein expression from such low-quality picture.
Thank you for your comments. As mentioned earlier, both CX3CR1 positive and negative Tregs reduce inflammatory proteins in the hippocampus caused by LPS, with CX3CR1 positive Tregs being more effective. We observed that control Tregs do not significantly affect COX2 levels. We regret the poor quality of the COX2 Western blot image. However, in the uncropped Western blot images provided, clearer bands can be observed, and since COX2 is a critical marker of inflammation, we request your understanding as we cannot readily exclude it from the manuscript.
Comments 5: 5B. IL-6 is the only cytokine showing a significant decrease CX3CR1+ dependent vs LPS in hippocampus. TNF-α decreased even in CX3CR1- animals compared to LPS and IL-1β does not show significant changes according to the treatment. The conclusion proposed is not supported by experimental results.
Thank you for your valuable feedback. In this study, we investigated the effects of Treg treatments on the expression of pro-inflammatory cytokines, including TNF-α, IL-6, and IL-1β, in an LPS-induced inflammation model. LPS, a potent activator of microglia, significantly increased the levels of TNF-α, IL-6, and IL-1β compared to the control group, highlighting its strong pro-inflammatory impact (Fig. 5B).
Beyond the general anti-inflammatory effects of Tregs, we sought to explore the unique properties of CX3CR1+ Tregs, which migrate effectively to inflammatory sites via the CX3CL1/CX3CR1 axis to suppress inflammatory responses. For TNF-α, CTRL Treg treatment alone resulted in a significant reduction, but CX3CR1+ Treg treatment demonstrated an even stronger anti-inflammatory effect, likely due to the greater number of cells migrating to the brain. In contrast, CTRL Tregs did not significantly affect IL-6 levels, whereas CX3CR1+ Treg treatment led to a marked reduction. These results indicate that the suppression of IL-6 may be significantly influenced by Treg concentration or specific mechanisms involved in their function. These results indicate that CX3CR1+ Tregs may be more effective than general Tregs in modulating certain pro-inflammatory cytokines, such as IL-6.
Comments 6: Fig.7 Not only CX3CR1+ Treg but also CTRL-Treg reduced Aβ plaques in cortex and hippocampus. The effect is not specific for CXCR1+, as above. Also, better quality representative figure of immunofluorescence stain would be.
Thank you for your feedback on Figure 7. As you mentioned, while Ctrl Tregs are less effective than CX3CR1+ Tregs, they also contribute to the reduction of Aβ plaques in both the cortex and hippocampus. However, significant effects on Iba-1 were observed only with CX3CR1+ Tregs. To address your feedback, we will improve the figure’s clarity and presentation by increasing the DPI, ensuring our results are communicated more effectively. We appreciate your thoughtful suggestions and understanding.
Comments 7: Figure 8. Suppression of Disease-Associated Microglial Activation by CX3CR1+ Tregs. Statements are not supported by enough data. Only CSF1 decreases in both cortex and hippocampus in CX3CR+Treg vs LPS. CX3CR1 decreased only in hippocampus between CX3CR1+ vs LPS. TMEM119-positive microglia did not consistently express polarized markers for anti-inflammatory or pro-inflammatory phenotype. So, it indicates an increase of microglia in cortex with no indication of polarization and thus function.
Thank you for your valuable feedback. In this analysis, CX3CR1 showed significant results in the hippocampus, and CSF1 was significantly reduced in both the cortex and hippocampus with CX3CR1+ Treg treatment. Unfortunately, we were unable to observe suppressive effects on other markers. TMEM119 is a microglial marker typically present in a healthy brain, but its expression tends to decrease in brain-injured mouse models. Interestingly, CX3CR1+ Tregs selectively increased TMEM119 expression in the cortex, a marker related to microglial phagocytic activity. This suggests that CX3CR1+ Tregs may play a crucial role in promoting the phagocytic function of microglia, potentially aiding in the removal of neurotoxic substances from the brain.
The corrected sentence will read:
(280-295) To evaluate activated microglia in 3xTg mice, we analyzed mRNA expression levels of various DAM markers in the cortex and hippocampus (Fig. 8A). CX3CR1 is an im-portant receptor that regulates neuron-microglia interactions during neuroinflammation, and CSF1 is a growth factor necessary for microglial proliferation and activation. Both of these markers showed increased expression in 3xTg mice, indicating enhanced microglial activation. CX3CR1+ Treg treatment suppressed this increased expression in the cortex and hippocampus, demonstrating a reduction in microglial activation. CCL6 and TREM2, which are associated with immune cell recruitment and neuroprotection respectively, as well as ITGAX, a marker involved in microglial activation, did not show significant results. TMEM119, a microglial marker present in a normal brain, tends to decrease in expression in mouse models with brain injury. Microglia do not simply increase inflammation; they also play a dual role by regulating inflammation through phagocytic activity (6). CX3CR1+ Tregs selectively increased TMEM119 expression in the cortex, which is associated with microglial phagocytic activity. This suggests that CX3CR1+ Tregs may play a significant role in promoting the phagocytic function of microglia, potentially aiding in the removal of neurotoxic substances from the brain (7).
Comments 8: ELISA data in Fig. 9B show TNF-α and IL-1β reduction in both ctrl and CX3CR1 Treg ns LPS with no specific effect for CX3CR1 Treg.
Thank you for your valuable feedback. As previously discussed, Tregs alone can elicit a response, as evidenced by the significant reduction in TNF-α and IL-1β levels in both ctrl Treg and CX3CR1 Treg groups compared to LPS. However, CX3CR1+ Tregs tend to be more effective in modulating the inflammatory response. Additionally, IL-1β and IL-23 showed significant results only in the cortex.

Reviewer 2 Report
Comments and Suggestions for Authors
Abstracting: The authors introduced CX3CR1+ Tregs into both acute and chronic neuroinflammation model mice and demonstrated that this approach suppressed inflammation. Data were comprehensively collected, spanning from molecular to animal levels. It is nearly acceptable.
1. Major Points
1.1. Purpose and Conclusion Correspondence
Fine.
1.2. Proper Introduction
- Please note that Iba1 is used as a microglial marker; this should be clarified initially.
- Please carefully introduce the genes shown in Fig. 8.
1.3. Figure Readability
Fine.
1.4. Figure Reliability
The Cx3cr1- and Cx3cr1+ groups in Figs. 3–9 might also have been treated with LPS or 3xTg, though this is not immediately clear. Please revise the figure legends to indicate treatments more explicitly, such as "LPS + Cx3cr1-" or "LPS + Cx3cr1+."
1.5. Missing Figures and Unreferenced Figures
In Fig. 8, ITGAX was not reffered. Fig.8 shows CCL6 but in maintext, it was written as CCL1. Please re-check.
1.6. Sufficient Explanation of the Results
When discussing the data from this study, it is recommended to include corresponding figure numbers, such as “~~~ (Fig. 3A)” for clarity, in Discussion section.
2. Minor Points
2.1. Typos
Please check superscript formatting, e.g., 10^5 should be corrected to 10510^5.
3. Reference Check
3.1. Percentage of Recent Works
Example: 6 out of 24 references (25%) are recent (after 2020). The manuscript lacks sufficient recent citations (<30%). Additionally, the total number of references is low. Please increase the citation count to at least 30.
3.2. Proper Referencing
Fine.
Author Response
Dear Reviewer,
Thank you for your valuable and insightful reviews. We deeply appreciate the opportunity to respond to your comments and incorporate your suggestions, which have allowed us to further improve the quality and precision of our manuscript.
We have provided detailed responses to each of your questions, with answers formatted in red for clarity, and the revisions are clearly indicated with underlines in the response letter. Changes within the manuscript are highlighted for ease of review. Additionally, to facilitate a better understanding of the experimental content, we have significantly revised the graphic abstract.
Abstracting: The authors introduced CX3CR1+ Tregs into both acute and chronic neuroinflammation model mice and demonstrated that this approach suppressed inflammation. Data were comprehensively collected, spanning from molecular to animal levels. It is nearly acceptable.
- Major Points
1.1. Purpose and Conclusion Correspondence
Fine.
1.2. Proper Introduction
- Please note that Iba1 is used as a microglial marker; this should be clarified initially.
Thank you for your detailed feedback. Iba-1 has been reported as a microglia-specific marker and is widely used for detecting microglia (1). We will add this information to the main text along with relevant references to clarify the experiment further.
The corrected sentence will read:
(157-158) Subsequently, we performed an analysis of activated microglia using Iba1 antibodies. Iba-1 has been reported as a microglia-specific marker and is widely used for detecting microglia (1).
- Please carefully introduce the genes shown in Fig. 8.
Thank you for your thoughtful feedback. We have added descriptions of the genes shown in Fig. 8 to further strengthen the results section.
The corrected sentence will read:
(280-295) To evaluate activated microglia in 3xTg mice, we analyzed mRNA expression levels of various DAM markers in the cortex and hippocampus (Fig. 8A). CX3CR1 is an im-portant receptor that regulates neuron-microglia interactions during neuroinflammation, and CSF1 is a growth factor necessary for microglial proliferation and activation. Both of these markers showed increased expression in 3xTg mice, indicating enhanced microglial activation. CX3CR1+ Treg treatment suppressed this increased expression in the cortex and hippocampus, demonstrating a reduction in microglial activation. CCL6 and TREM2, which are associated with immune cell recruitment and neuroprotection respectively, as well as ITGAX, a marker involved in microglial activation, did not show significant re-sults. TMEM119, a microglial marker present in a normal brain, tends to decrease in expression in mouse models with brain injury. Microglia do not simply increase inflammation; they also play a dual role by regulating inflammation through phagocytic activity (2). CX3CR1+ Tregs selectively increased TMEM119 expression in the cortex, which is associ-ated with microglial phagocytic activity. This suggests that CX3CR1+ Tregs may play a significant role in promoting the phagocytic function of microglia, potentially aiding in the removal of neurotoxic substances from the brain (3).
1.3. Figure Readability
Fine.
1.4. Figure Reliability
The Cx3cr1- and Cx3cr1+ groups in Figs. 3–9 might also have been treated with LPS or 3xTg, though this is not immediately clear. Please revise the figure legends to indicate treatments more explicitly, such as "LPS + Cx3cr1-" or "LPS + Cx3cr1+."
Thank you for your careful feedback. To improve clarity, we will revise each figure legend to specify all treatment conditions more clearly, using labels such as "WT-PBS," "LPS-PBS," "LPS-ctrl Treg," "LPS-CX3CR1 Treg," "3xTg-PBS," "3xTg-ctrl Treg," and "3xTg-CX3CR1 Treg." This adjustment will ensure that readers can easily understand the specific treatment conditions across Figures 3–9.
1.5. Missing Figures and Unreferenced Figures
In Fig. 8, ITGAX was not reffered. Fig.8 shows CCL6 but in maintext, it was written as CCL1. Please re-check.
Thank you for your valuable feedback. We have added ITGAX to the main text and corrected the typographical error regarding CCL1. The revised content is clearly referenced in Section 1.2.2 above.
1.6. Sufficient Explanation of the Results
When discussing the data from this study, it is recommended to include corresponding figure numbers, such as “~~~ (Fig. 3A)” for clarity, in Discussion section.
Thank you for your thorough review. As per your suggestion, we will revise the Discussion section to include relevant figure numbers.
- Minor Points
2.1. Typos
Please check superscript formatting, e.g., 10^5 should be corrected to 10510^5105.
Thank you for your detailed review. We have corrected all formatting errors in the text to display 105 properly.
- Reference Check
3.1. Percentage of Recent Works
Example: 6 out of 24 references (25%) are recent (after 2020). The manuscript lacks sufficient recent citations (<30%). Additionally, the total number of references is low. Please increase the citation count to at least 30.
Thank you for the suggestion. We will add more recent references to increase the total number of citations to 30.
3.2. Proper Referencing
Fine.
- Korzhevskii DE, Kirik OV. Brain Microglia and Microglial Markers. Neuroscience and Behavioral Physiology (2016) 46(3):284-90. doi: 10.1007/s11055-016-0231-z.
- Colonna M, Butovsky O. Microglia Function in the Central Nervous System During Health and Neurodegeneration. Annu Rev Immunol (2017) 35:441-68. Epub 20170209. doi: 10.1146/annurev-immunol-051116-052358.
- Mercurio D, Fumagalli S, Schafer MK-H, Pedragosa J, Ngassam LDC, Wilhelmi V, et al. Protein Expression of the Microglial Marker Tmem119 Decreases in Association with Morphological Changes and Location in a Mouse Model of Traumatic Brain Injury. Frontiers in cellular neuroscience (2022) 16. doi: 10.3389/fncel.2022.820127.
